# Complete Chloroplast Genome of *Rhipsalis baccifera,* the only Cactus with Natural Distribution in the Old World: Genome Rearrangement, Intron Gain and Loss, and Implications for Phylogenetic Studies

**DOI:** 10.3390/plants9080979

**Published:** 2020-07-31

**Authors:** Millicent Akinyi Oulo, Jia-Xin Yang, Xiang Dong, Vincent Okelo Wanga, Elijah Mbandi Mkala, Jacinta Ndunge Munyao, Victor Omondi Onjolo, Peninah Cheptoo Rono, Guang-Wan Hu, Qing-Feng Wang

**Affiliations:** 1CAS Key Laboratory of Plant Germplasm Enhancement and Specialty Agriculture, Wuhan Botanical Garden, Chinese Academy of Sciences, Wuhan 430074, China; millicentoulo@gmail.com (M.A.O.); yangjxgz@163.com (J.-X.Y.); directx0831@163.com (X.D.); vincentokelo@gmail.com (V.O.W.); mkala@wbgcas.cn (E.M.M.); jacintandunge.jn@gmail.com (J.N.M.); Onjolovictor@gmail.com (V.O.O.); Peninahrono@yahoo.com (P.C.R.); qfwang@wbgcas.cn (Q.-F.W.); 2Sino-Africa Joint Research Center, Chinese Academy of Sciences, Wuhan 430074, China; 3University of Chinese Academy of Sciences, Beijing 100049, China

**Keywords:** *Rhipsalis baccifera*, Cactaceae, chloroplast genome, rearrangements, phylogeny

## Abstract

*Rhipsalis baccifera* is the only cactus that naturally occurs in both the New World and the Old World, and has thus drawn the attention of most researchers. The complete chloroplast (cp) genome of *R. baccifera* is reported here for the first time. The cp genome of *R. baccifera* has 122, 333 base pairs (bp), with a large single-copy (LSC) region (81,459 bp), SSC (23,531 bp) and two inverted repeat (IR) regions each 8530 bp. The genome contains 110 genes, with 73 protein-coding genes, 31 tRNAs, 4 rRNAs and 2 pseudogenes. Twelve genes have introns, with loss of introns being observed in, *rpoc1*
*clpP* and *rps12* genes. 49 repeat sequences and 62 simple sequence repeats (SSRs) were found in the genome. Comparative analysis with eight species of the ACPT (Anacampserotaceae, Cactaceae, Portulacaceae, and Talinaceae) clade of the suborder Portulacineae species, showed that *R. baccifera* genome has higher number of rearrangements, with a 19 gene inversion in its LSC region representing the most significant structural change in terms of its size. Inversion of the SSC region seems common in subfamily Cactoideae, and another 6 kb gene inversion between *rbcL- trnM* was observed in *R. baccifera* and *Carnegiea gigantea*. The IRs of *R. baccifera* are contracted. The phylogenetic analysis among 36 complete chloroplast genomes of Caryophyllales species and two outgroup species supported monophyly of the families of the ACPT clade. *R. baccifera* occupied a basal position of the family Cactaceae clade in the tree. A high number of rearrangements in this cp genome suggests a larger number mutation events in the history of evolution of *R. baccifera*. These results provide important tools for future work on *R. baccifera* and in the evolutionary studies of the suborder Portulacineae.

## 1. Introduction

Cactaceae is one of the most conspicuous and diverse angiosperm families of warm arid America [1]. This family is monophyletic [2] and belongs in the suborder Portulacineae of the order Caryophyllales [3]. In this order, family Cactaceae forms a unique clade called ACPT clade together with genus *Talinum*, genus *Portulaca* and family Anacampseroteae [4]. The basal position of this clade is occupied by genus *Talinum* which forms the sister group to a subclade consisting of family Anacampseroteae, family Cactaceae, and genus *Portulaca*. The relationship of the members of this clade, however, remains unclear [4,5,6,7]. Family Cactaceae is almost endemic to the New World except for the epiphytic *Rhipsalis baccifera* (J.S.Muell.) Stearn, which is found naturally occurring in both the New and the Old Worlds [8]. 

*Rhipsalis baccifera* (J.S.Muell.) Stearn, commonly known as the mistletoe cactus, belongs to genus *Rhipsalis* tribe *Rhipsalideae* of the subfamily Cactoideae [9,10]. This species is morphologically different from its putative terrestrial Cactoideae ancestors, mainly due to the presence of dispersed areoles with minute, bristly spine-like structures (as opposed to large sclerified spines) and its pendulous epiphytic lifestyle in the humid tropics [11]. It being the only cactus species naturally occurring outside the New World [8], has drawn the attention of many scientists. However, despite several theories being proposed to explain the distribution of *R. baccifera,* it is still unclear why it is the only Cactaceae species that occur naturally outside the New World [1,12,13]. *R. baccifera* produces soft, small (5–8 mm in diameter) globose, sweet and juicy fruit that resemble small grapes, which are fed on by both birds and human beings [14]. It is of medical importance, in that its stem is crushed and used with the juice of *Lonchocarpus chrysophyllus* Kleinhoonte to treat the bites of coral snakes (*Micrurus* sp.) and also used together with *Philodendron* sp. to soothe the wounds of venomous stingrays (*Potamotrygon* sp.) [15]. Additionally, the whole plant is used as an ingredient in a curative herbal bath [2]. *R. baccifera* is currently described as having six [9]. The six subspecies include; *R. baccifera* subsp. *horrida* (Baker) Barthlott, *R. baccifera* subsp. *baccifera*, *R. baccifera* subsp. *erythrocarpa* (K. Schumann) Barthlott, *R. baccifera* subsp. *mauritiana* (De Candolle) Barthlott, *R. baccifera* subsp. *shaferi* (Britton and Rose) Barthlott and N.P. Taylor and *R. baccifera* subsp. *hileiabaiana* (J.L. Hage and H.S. Brito) N.P. Taylor and Barthlott [9]. Furthermore, *R. baccifera* has undergone successive polyploidization events and its chromosome number varies from diploid (2n = 2x = 22) to tetraploid (2n = 4x = 44) to octaploid (2n = 8x = 88) [9]. Higher levels of polyploidy are correlated with increased geographic distance from Brazil which is its centre of diversity [16]. 

Previous studies regarding *Rhipsalis baccifera* ranged from the quest to understand its distribution [9,12], morphological characters [11,17], and molecular systematics [13,18]. One of the most comprehensive works so far on the evolutionary analysis of *R. baccifera* proposed that this species might have undergone a longer history of evolution in comparison to other cacti but suggested more work to support this idea [19]. Despite the increased attention from researchers, evolutionary history of *R. baccifera* remains unclear. Thus, an analysis of complete chloroplast genome of *R. baccifera* will be of high significance to shed more light on its evolutionary history. 

Chloroplast genomes have low evolution rate and maternal inheritance [20] and have thus been found to be ideal in studies of plant phylogeography and molecular evolution, as well as phylogenetic analysis [21,22,23,24]. Despite cp genomes of vascular plants being highly conserved in their basic structures and their rates of nucleotide substitutions being generally slow, comparative genomic studies in the past have revealed occasional structural changes, such as inversions, gene or intron losses, and rearrangements among plant lineages [25,26,27,28]. Rearrangements in the chloroplast genome were considered to have occurred rarely enough in evolution that they can be used to delineate major groups [29]. Plastid genomes have played a key role in the evolutionary and phylogenetic studies due to their small size and highly conserved gene content and order. Therefore, in this study, we compare the cp genome of *R. baccifera* to cp genomes of seven other genomes of the ACPT clade to study their relationship. They include; *Talinum paniculatum, Portulaca oleraceae, Carnegieae gigantea, Lophocerus schottii, Mammilaria albiflora, M. solisioides* and *M. zephyranthoides* (the three species of genus *Mammillaria* represent the three cp genome structures (1–3) respectively found in its members). So far, these are the only available complete cp genomes of members of this clade, in addition to the other four species of genus *Mammillaria* previously studied [30] but not mentioned here. *Spinacia oleraceae* L. was also used in this study as a representative genome of order Caryophyllales. Thus, this study aims at, (i) generating the first complete chloroplast analysis study of *R. baccifera,* (ii) conducting complete comparative genomics against other cp genomes of selected members of the ACPT clade of suborder Portulacineae and, (iii) providing insights into the phylogenetic relationships between members of the family Cactaceae, and the ACPT clade. 

Incongruences between biological and legal criteria for determining sub specific taxa and the continuing debate among taxonomists regarding the validity of subspecies as a taxonomic unit have resulted in controversies when listing sub species and thus more work is needed to support sub specific nomino [31,32]. Moreover, *R. baccifera* is recorded under its species name both in the Flora of Madagascar and the Flora of Tropical East Africa, though the species have a few differences in their morphology, suggesting non-recognition of the sub specific classification. We therefore choose to retain the species name in our study.

## 2. Results

### 2.1. Chloroplast Genome Organization and Features of Rhipsalis baccifera

The Complete chloroplast genome of *R. baccifera* has a total length of 122,333 bp in size. It displays a typical quadripartite structure with a large single-copy (LSC) region of 81,459 bp in length, separated from the SSC region of 23,531 bp, by two inverted repeat (IR) regions both 8530 bp long (Figure 1, Table 1).

The SSC region has higher GC content (39.3%) compared to the LSC region (36.0%) and the IR (36.9%) There are 110 unique genes in the cp genome of *R. baccifera*. Among the 110 genes, 73 are protein-coding genes, 31 *tRNA* genes and four *rRNA* genes (Figure 1; Table 1). Two pseudogenes, i.e., *ycf1* and *rpl23*, were detected in the cp genome of *R. baccifera*. A total of 12 genes have introns, with *ycf3* having two introns and the others *(trnI-GAU, trnA-UGC, trnL-UAA, trnG-UCC, trnK-UUU, rpL16, petB, petD, rps12, atpF, rps16)* having one intron. The *rps12* gene is transplicated but unlike in the chloroplast genomes of most species with three exons, where the first exon is located in the LSC region, while the other two exons are duplicated in IR regions, in *R. baccifera* cp genome, *rps12* gene has two exons with the first exon located in the LSC and the second exon in the SSC region. Pseudogenization of genes has also been observed in *rpl23* and *ycf1.* Additionally, the *ndhD* gene is a pseudogene in *Carnegiea gigantea, Lophocereus schottii* and genus *Mammillaria*, while this is not the case in *R. baccifera* and *T. paniculatum*. Additionally, the *ndhJ* gene, which might have been lost in the chloroplast genomes of the so far analyzed Cactaceae species, i.e., *Lophocereus schottii, Carnegiea gigantea* and in members of genus *Mammillaria*, was discovered in the chloroplast genome of *R. baccifera.* It is found located between *trnD-GUC* and *trnF-GAA*. However, this gene also exists in the cp genomes of *Talinum paniculatum* and *Portulaca oleracea*, two members of this suborder Portulacineae, though at a different location; between gene *ndhK* and *trnF-GAA*.

Furthermore, the cp genome of *R. baccifera* displayed additional features that were absent in the cp genomes of other cacti, but present in the cp genomes of *Talinum paniculatum* and *Portulaca oleracea*. *R. baccifera* might have gained introns or maybe retained them in some genes. The *petB* gene in *R. baccifera* cp genome for example, has an intron as is the case in *Portulaca oleracea* and *T. paniculatum*, whereas this gene lacks introns in most cp genome of members of Cactaceae family so far studied.

### 2.2. Inversions and Rearrangements in the cp Genome of Rhipsalis baccifera

#### 2.2.1. *rbcL - atpB - atpE-trnM* Inversion

The gene order in the ancestral large single copy (LSC) region has been partially retained in the cp genome of *R. baccifera* from the ideal order across angiosperms with some few inversions observed; for example, a small ~6 kb inversion involving four genes: *rbcL - atpB - atpE-trnM* (Figure 2). *R. baccifera* and *C. giganteae* have lost the *trnV* gene. 

#### 2.2.2. A 19-Gene Inversion in the LSC of *Rhipsalis baccifera*

The LSC region of *R. baccifera* also displays a large inversion of 19 genes (Figure 3). The region occupied by genes *ndhJ-trnF-trnL-trnT-rps4-trnS-ycf3-psaA-psaB-rps14-trnfM-trnG-psbZ-trnS-psbC-psbD-trnT-trnE-trnY* in the chloroplast genome of *R. baccifera* has been inverted. *R. baccifera* cp genome was only compared to that of *S. oleraceae* which is a representative genome of order Caryophyllales since this inversion has not been previously observed in other Caryophyllales so far studied.

#### 2.2.3. Rearrangements in the SSC Region

The IR region of the *R. baccifera* cp genome is significantly contracted and hence some genes that are ideally located in the IR regions of most angiosperms cp genomes have been moved to the SSC region (Figure 1). Moreover, we observed an inversion of the SSC region of the chloroplast genome of *R. baccifera* compared to the structure of other cp genomes (Figure 4). We compared *R. baccifera* to *T. paniculatum* since *C. gigantea* has lost its IRs, and most species of genus *Mammillaria* except *M. zephyranthoides* have IRs that are highly contracted. The arrangement of these genes is similar in *T. paniculatum* and *P. oleraceae,* which are similar to that of most angiosperms. 

### 2.3. Amino Acid Composition of Coded Proteins in R. baccifera cp Genome

According to the analysis of codon usage in this study, 20,555 codons are involved in coding of 73 protein genes and 31 tRNA genes in the cp genome of *Rhipsalis baccifera*. Leucine (10.14%) and Isoleucine (7.88%) were the most frequently used amino acids, while codons for cysteine (1.08%), represented the least prevalent amino acids (Figure 5a. This has been also observed for other members of Cactaceae family analyzed in this study. UAA was also observed to be the most frequently used stop codon in *R. baccifera* (Appendix A). Codon usage in the suborder Portulacineae is generally conserved, with no significant variation (Figure 5b). 

### 2.4. Repeats

A total of 49 repeat sequences were identified using REPuter software. Out of which, 33 were forward repeats and 16 were palindromic repeats. No complement and reverse repeats were found in the chloroplast genome of *R. baccifera*. The repeats ranged from 46 to 122 bp in length. The majority were located in the intergenic spacer (IGS) and intron sequences. Twenty repeats are located in exons of *rps7, rps19, rps18, accD, clpP and ycf1* (Appendix A).

Another type of repeat sequence found in chloroplast genomes is the simple sequence repeats (SSRs), also called microsatellites. In the cp genome of *R. baccifera,* there are 62 SSRs, with the highest number being of mononucleotide repeats (38). There were also 9 dinucleotide repeats, 5 trinucleotide repeats, 8 Penta nucleotide repeats, and 2 hexanucleotide repeats (Figure 6a). Compared to other members of suborder Portulacineae, *Portulaca oleraceae* had the highest number of SSRs (111), while *Carnegiea gigantea* had the least (43) (Figure 6b) (S4). 

### 2.5. Boundaries between IRs and the Single Copy Regions

The contraction or expansion of the IR regions varies in the cp genomes of various plants. In this study, we analyzed the IR contraction and expansion of the *R. baccifera* cp genome by comparing the IR-LSC and IR-SSC borders with those of four other species of suborder Portulacineae. For the family Cactaceae *Mammillaria solisioides* was randomly selected to represent the genus *Mammillaria*, whereas *Carnegiea gigantea,* though having lost one of its IR regions, was also used as in [33]. Among them, *Talinum paniculatum* had the longest chloroplast genome, while *C. gigantea* had the shortest. Rearrangement was observed in the SSC region of family Cactaceae. In *M*. *solisioides*, *ycf2* stretches out of the IR region due to the expansion of the IR and rearrangement within its cp genome. The IR region of *R. baccifera* was observed to be highly contracted, resulting in the pseudogene of *rps23* being located in the LSC region instead of the IR. *ycf1* gene in *R. baccifera* is s pseudogenized (Figure 7).

### 2.6. Genome Comparison and Sequence Divergence

The overall sequence identity of the complete cp genome sequences of eight species of suborder Portulacineae, including *R. baccifera* was comparatively analyzed using MVISTA software with annotation of *R. baccifera* as the reference (Figure 8). The results show that the sequences vary in length with *Talinum paniculatum* being the longest (156,929 bp), and *Mammillaria zephyranthoides* the shortest (107,343 bp). Compared to the non-coding regions, the coding regions seemed more conserved. The LSC regions are generally least conserved among these species of suborder Portulacineae with notable divergences in some genes such as *rps*16, *accD, ndhJ and ndhK.*

MAUVE graphic of the structural alignments of complete chloroplast genomes of these species also revealed divergences. We first compared species of family Cactaceae (Figure 9), where a unique inversion and rearrangement was observed only in *R. baccifera* at the LSC region from gene *psaI*–*trnY* which is flipped in the opposite direction compared to their arrangements in the cp genomes of the other species. When compared to other species of suborder Portulacineae (Appendix A), this rearrangement was still only unique to *R. baccifera*. Inversions previously observed in the LSC region *R. baccifera* are also evident in the structural alignment.

Moreover, in genus *Mammillaria,* an inversion and rearrangement were observed in the LSC at the region between genes *trnM-CAU and trnE-UUC* (highlighted in red). In addition, there is a rearrangement in the cp genome structure of *M. zephyranthoides* (circled) (Figure 9) below.

### 2.7. Phylogenetic Analysis

The phylogenetic analysis involved 36 species of order Caryophyllales and two outgroups. The species of suborder Portulacineae form a clade supported by strong bootstrap values, within which a clade of family Cactaceae is displayed (Figure 10). Interestingly, *R. baccifera* forms a branch occupying a basal position in the clade formed by Cactaceae species. This can be used in future studies of this species to understand its relationship to other Cacti and its natural occurrence outside the New world (Appendix A) (Figure 10).

## 3. Discussion

### 3.1. Chloroplast Genome Organization and Features of Rhipsalis baccifera

The chloroplast genome of *R. baccifera* displays a typical angiosperm cp genome structure [34] of a quadripartite organization consisting of two copies of inverted repeats (IRs), a large single copy (LSC) region and a small single copy region. However, this cp genome also displays some features that are uncommon in other cp genome structures.

The inverted repeat regions (IRs) are contracted and have consequentially resulted in some typical IR region genes to be located in the LSC and SSC regions. IR contraction has been previously observed in other species and described to be as a result of evolution [35,36]. Some genes in *R. baccifera* also displayed some interesting features. Loss of introns for example, was observed in some genes. The *rps12* gene in this cp genome has lost one intron. This loss of the *rps*12 3’end intron has also been observed in Aspagarales [37] and is described to be as a result of convergent evolution. The loss of an intron of *rps12* gene has also previously been observed in legumes [38] and in some parasitic plants such as *Cuscuta* [39]. However, in comparison to other species of the suborder Portulacineae, *Mammillaria zephyranthoides*, *Talinum paniculatum* and *Portulaca oleraceae* displayed the common 3 exons feature of the *rps12* gene, while other members of genus *Mammillaria* had three exons of the *rps12* gene but two exons are in the LSC region and one in the SSC, and this may be due to the IR contraction observed in these species [30,33,40]. *Carnegiea gigantea*, despite losing its IRs, also still retains three exons of the *rps12* gene, by displaying a three-part structure: two copies of 5’ exon in inverted orientation and one copy of 3’ exon [41]. *rps12* gene is generally highly conserved and changes in its structure have been previously suggested to be a result of evolution [42,43]. The loss of introns in the *clpP* gene and has only been observed in the cp genome of *R. baccifera* among all species of suborder Portulacineae studied here. However, this has previously been observed in the chloroplast genomes of some species of order Caryophyllales, specifically in tribe *Sileneae* [44,45]. Interestingly, according to a previous study, the inversion of *trnY-psaI* region is always coupled with the loss of introns in the *clpP* gene, which has also been observed in *R. baccifera.* This feature has been found in the species of tribe *Sileneae* that display the same unusual trait in their cp genomes e.g., *Silene chalcedonica* and *S. notiflora* [45]. Assuming this is not ancestral, as it is not found in the most common chloroplast genomes of Caryophyllales, this thus indicates that *R.baccifera* has undergone a number unusual changes in the course of its evolution, which though uncommon, are similar to those in some species of tribe Sileneae. The loss of both *clpP* gene and *rps12* gene in the same chloroplast genome was first observed in, *Cicer arietinum* [38] and was described as unique. The *petB* gene in *R. baccifera* cp genome has an intron as is the case in *P. oleracea* and *T. paniculatum*, whereas this gene lacks introns in most cp genome of members of Cactaceae family so far studied, but was observed in a recent study to be present in *Opuntia quimilo* [46]. Assuming *R. baccifera* has gained the *petB* gene intron, then *R. baccifera* might have gained this intron during its evolution to enhance the expression level of this *petB* gene. Introns are important in the regulation of gene expression. They can enhance the gene expression level, on the special position, in the specific time [24]. 

Additionally, the contraction of the IR region led to the pseudogenization of the *rpl23* gene and its position in the LSC region of *R. baccifera* cp genome unlike in other angiosperms where it is found in the IRs.

The *ndhJ* gene is important for the stability of the activities of NDH-1 complexes that are essential components in PSI cyclic electron flow during photosynthesis [47]. The gene, which might have been lost in the chloroplast genomes of the so far analyzed Cactaceae species i.e., *L. schottii, C. gigantea* and in members of genus *Mammillaria* [40,41], was discovered in the chloroplast genome of *R. baccifera.* It is located between *trnD-GUC* and *trnF-GAA*. The existence of *ndhJ* gene has also been observed in the cp genomes of other Caryophyllales, e.g., in some members of suborder Portulacineae [33,40], though at differing locations. For instance in *T. paniculatum* and *P. oleracea*, two members of this suborder, the gene is found between *ndhK* and *trnF-GAA*. The difference in location is probably because of the inversion in the LSC region of *R. baccifera* affecting genes *ndhJ-trnY.*


Gene pseudogenization is observed to differ in these genomes. The *ycf1* gene for example, is pseudogenized in the cp genome of *R. baccifera*, while this is not the case in other previously studied Cacti [30,41]. Additionally, the *ndhD* gene is a pseudogene in *C. gigantea, L. schottii* and genus *Mammillaria*, while this is not the case in *R. baccifera* and *T. paniculatum*. The *rpl23* in *R. baccifera* is a pseudogene as in other cacti, but unlike other members of family Cactaceae, where it is located in the IR region e.g., in genus *Mammillaria* [30], for the case of *R. baccifera*, this pseudogene is found in the LSC region and this might be as a result of IR contraction in its cp genome.

### 3.2. Inversions and Rearrangements in the cp Genome of Rhipsalis baccifera

The structure of the *R. baccifera* cp genome also displays several inversions, gene losses, and transpositions in the LSC region. There is a small ~6 kb inversion involving four genes: *rbcL - atpB - atpE-trnM* (Figure 2). This has also been previously observed in *Carnegiea gigantea* [41] and other Caryophyllales such as *Atriplex hastata, Chenopodium murale* and *Pereskia grandiflora* and in *Lychnis wilfordii* [44,48], though as in *Carnegiea gigantea*, this inversion also led to the loss of *trnV* gene in *R. baccifera*. However, in all these other species, this inversion seems to have occurred once, but in *R. baccifera* cp genome, the *rbcL* gene seems to have been inverted again. Though the inversion of the *rbcL - atpB - atpE-trnM* region seems to be a common feature among members of order Caryophyllales, the second time inversion of the *rbcL* gene seems to be unique to only *R. baccifera*. We suppose that this might be because *R. baccifera* might have undergone more mutational events during its evolution. However, more work should be done to clarify this.

A larger inversion was additionally observed in the LSC region of *Rhipsalis baccifera* that constituted an inversion of 19 genes. The LSC region occupied by genes *ndhJ-trnF-trnL-trnT-rps4-trnS-ycf3-psaA-psaB-rps14-trnfM-trnG-psbZ-trnS-psbC-psbD-trnT-trnE-trnY* in the chloroplast genome of *R. baccifera* has been inverted and this has not been observed in other Cactaceae species or suborder Portulacineae species previously studied. However, similar inversions have been partially observed in two other species of the order Caryophyllales; *Silene noctiflora* and *Mammillaria zephyranthoides. S. noctiflora* was previously described as having the most complicated structure of the cp genome in order Caryophyllales due to a high number of inversions, rearrangements and transpositions that also occurred in this region for genes *psbD*-*accD*, *petL-clpP, trnD-T* and *psaI-psbE* [45]. In *M. zephyranthoides,* the LSC region seems to have undergone several inversions and transpositions too [30]. The inversions and transposition events seem to have occurred several times and we suggest that first, there was an inversion and transposition of the region *psaI-petA*, then the transposition of the *accD-trnM,* followed by another inversion of *accD* gene, and later the *rbcL-trnM* inversion. However, the inversions at the LSC region of *R. baccifera* seems to have been larger compared to *M. zephyranthoides*. Our study therefore provides important insights for future studies on the relationship between these species. 

Moreover, we observed another inversion of the SSC region of *R. baccifera* chloroplast genome in comparison to the structure of most angiosperms’ cp genomes. This was also observed in the cp genome of *C. gigantea* and *M. solisioides* but was not the case in *Opuntia* as in a recent study of *O. quimilo* [46]. We suggest that this might be a unique feature of subfamily Cactoideae.

### 3.3. Amino Acid Composition of Coded Proteins in R. baccifera

The genetic codes of different organisms are often biased toward using one of the several codons that encode the same amino acid over others [49]. Changes in the codon usage tend to reflect the evolution of chloroplasts [50], and thus might provide insight on the evolutionary history is a measure of non-uniform synonymous codon usage in coding sequences. It is the ratio between frequency of use and expected frequency of a particular codon. Relative synonymous codon usage (RSCU) values <1.00 indicate use of a codon less frequent than expected, while codons used more frequently than expected have a score of >1.00 [51]. 

This study concluded that the total number of codons that are involved in protein coding in the cp genome of *R. baccifera* is 20,555 codons describing the coding capacity of the 73 protein-coding genes and 31 tRNA genes of *R. baccifera* chloroplast genome. Codon usage in the suborder Portulacineae is generally conserved, with low variation.

### 3.4. Repeats

Repeat sequences provide important information about genomes. The highly rearranged plastomes contain a high frequency of large repeats [52], some of which represent full or partial duplications of genes. Repetitive DNA in the chloroplast genome of *R. baccifera* may be associated with the elongations of protein-coding genes and pseudogenes (e.g., *rps23, ycf1,* and *rps19*), *tRNA* genes (e.g., *trnL-CAA* and *trnfM-CAU*), and intergenic spacers of both single copy and IR regions. *R. baccifera* has a higher frequency of large repeats among these species and thus also the most rearranged. The exon of the *accD* gene has a high frequency of repeats that range between 48–82 bp, which we hereby suggest as a result of gene elongation as also witnessed by the larger size of this gene in *R. baccifera* cp genome in comparison to other Portulacineae cp genomes. This is also the case in the *rps18* gene of this genome which we also assume might have undergone gene elongation. 

Microsatellites are tandem repetitive DNA sequences, comprising of one to six (mono-, di-, tri-, tetra-, penta-, and hexa-) repeat nucleotide units. Their highly polymorphic nature makes them important molecular markers in plant systematics, in mapping genomes, in population structure and evolutionary processes, taxonomic studies among others [53]. In the cp genome of *R. baccifera,* there are 62 SSRs; with the highest number being of mononucleotide repeats (38) (Figure 6b). Mononucleotide SSRs were the most abundant repeats in all species, with A/T mononucleotide repeats being the most frequent SSRs. This result is consistent with previous studies that A/T repeats are the most abundant SSRs in the cp genomes [33]. These identified repeats can be useful in future population genetics and phylogenetic studies involving *R. baccifera.*

### 3.5. Boundaries between IRs and the Single Copy Regions

Changes in the IR region such as expansion, contraction or even loss are one of the common types of gene order rearrangement types that occur in the chloroplast genomes, and are presumed to might have happened during genome evolution. For example, both the loss of one copy of the IR and inversions are extremely useful characters in legume phylogeny [54,55], defining large clades within the family. 

In the chloroplast genome of *R. baccifera*, the boundaries of the IR and Single Copy (SC) regions suggest a contraction of the IR regions of *R. baccifera* cp genome. Compared to the typical structure of the IR region in the cp genome of tobacco, the typical angiosperm cp genome structure [56], the IR regions in *R. baccifera* have been contracted resulting in the displacement of most usually IR-located genes to the SC regions. This has also resulted in the loss of a copy of RNA genes that are normally duplicated in the IRs as seen in Figure 7. The contraction of the IR as explained before, has resulted in a high number of repeats in the IR regions and the intergenic spacer regions found in the IR borders. IR contraction observed in the cp genome of *R. baccifera* might be a result of rearrangements and structural changes that occurred during its genome evolution. *R. baccifera* might have also lost the PPR4 (pentatricopeptide repeat (PPR), which is a degenerate 35–amino acid repeat motif that is widely distributed among eukaryotes) which is associated in vivo with *rps12-intron 1* and is also required for its splicing [57].

### 3.6. Genome Comparison and Sequence Divergence

MVISTA analysis reveals that LSC regions are generally least conserved in suborder Portulacineae with notable divergences in some genes such as *rps*16, *accD, ndhJ and ndhK.* The *accD* gene in *R. baccifera* is the largest in size (3072 bp) in all the analyzed representatives of suborder Portulacineae herein. It is also pseudogenized in all other species of family Cactaceae analyzed herein. High divergence was observed in genes *ndhK, ndhJ and rps16*. Genes *ndhK* and *ndhJ* are absent in the chloroplast genomes of the so far analyzed Cactaceae species except in *R. baccifera,* while the gene *rps*16 is pseudogenized in genus *Mammillaria*. The pseudogenization of the *rps16* gene has also been observed in other studies and has been assumed to be as a result of the dual targeting of the nuclear *rps16* copy to the plastid and mitochondria, signifying that the chloroplast-encoded *rps16* gene, which is important for plant survival, has been functionally replaced by a nuclear gene that can encode both mitochondrial and chloroplast thus the chloroplast gene *rps16* has been silenced and has become a pseudogene and its function replaced by the nuclear-encoded *rps16* [58].The *clpP* gene in *R.baccifera* has also lost introns. The *ycf2* gene is also pseudogenized in *M. zephyranthoides* and reduced in size unlike in other species of Portulacineae.

Moreover, the comparative analysis using MAUVE reveals inversions in *R. baccifera,* in the LSC regions; (1) *rbcL-trnM* and (2) *ndhJ-trnY* (which might be the unique feature of this genome) and (3) a transposition and loss of introns of gene *clpP* and (4) loss of one intron of *rps12* gene. *M. zephyranthoides* has two rearrangements in addition those observed in other species of this genus i.e., (1) region between genes *rps18-trnW* and (2) *psbZ-trnT* genes. We therefore conclude that these multiple inversions, intron loss and rearrangements observed in these species is as a result of parallel evolution in this order, and we agree with previous studies which suggested that there is a correlation between gene rearrangements, and changes in the genome structure of a chloroplast genome of a species and evolution [52].

### 3.7. Phylogenetic Analysis 

The combined Bayesian/maximum likelihood tree showed that *R. baccifera* formed a subclade from other members of family Cactaceae, which occupied a basal position of the family Cactaceae clade in the tree, with strong support values. Chloroplast genome rearrangements have been previously described as mutational events [29] and we thus propose that this species might have undergone a larger number of mutational events during its evolution in comparison to other species of family Cactaceae. We also suggest that these structural changes in the cp genome of *R. baccifera* might have occurred after *R. baccifera* had diverged from other cacti, which could explain the basal position of this species when phylogeny is inferred based on chloroplast sequences. 

Therefore, in regard to the explanation of the unique distribution of *R. baccifera,* our study rules out the proposed idea of *R. baccifera* being a primitive member of family Cactaceae and we support the idea of bird dispersal proposed by Barthlott [12] and further redefined by Korotkova [19] as having occurred several times to and from the New and Old Worlds. 

The members of the ACPT clade herein also form a common clade. The relationship between the members of this clade has been previously described as unclear [4]. However, few studies have been done using chloroplast DNA to analyze their relationship. This study involves the use of complete chloroplast genomes to analyze their phylogenetic relationship, and though not all representatives were available in our analyses, Figure 10, suggests a close relationship between *Talinum paniculatum* and *Portulaca oleraceae* as suggested by previous studies [7,59,60], therefore differing from conclusions of previous studies [33]. We hereby suggest that more work should be done on the phylogenetic analysis of these species using their complete chloroplast genomes.

## 4. Materials and Methods

### 4.1. Chloroplast Genome Sequencing and Assembly 

Fresh *Rhipsalis baccifera* leaves were collected from Kasigau forest in Taita (03°50’S, 38°39’E, Alt. 800 m), Kenya, and fast dried in silica gel for DNA extraction. The dried leaves and voucher specimens were later stored at the East Africa Herbarium, National Museums of Kenya (EA) and the Herbarium of Wuhan Botanical Garden (HIB) (Voucher Number: SAJIT- 3421). The genomic DNA was extracted from about 100 micrograms of the leaves using a modified cetyltrimethylammonium bromide (CTAB) method as described by Doyle [61].

The genome was sequenced using the Illumina platform at Novo gene Company (Beijing, China). The low quality data and adaptors were later filtered, and the obtained clean data was assembled using GetOrganelle-1.6.2 software [62], with *Arabidopsis thaliana* as the reference genome. The assembled genome was then manually corrected. We used Bandage software to check these results of the assembled genome and we then selected an optimal result. Later, the large reverse areas in these results, were manually adjusted. Finally, Geneious Prime 2019.2.1 (https://www.geneious.com) was used to determine the inverted repeat regions. 

### 4.2. Genome Annotation and Codon Usage Analysis

GeSeq online tool was then used to annotate the assembled chloroplast genome with default settings [63]. Annotations of tRNAs were confirmed using the tRNAscan-SE [64]. The validation of the cp genome of *R. baccifera* was then done using a standalone command line annotation tool called PGA (Plastid Genome Annotator) [65]. The gene map of the complete cp genome was drawn using OrganellarGenome DRAW software [66] (Figure 1). The annotated cp genome was submitted to GenBank (GenBank number: MT821847).

MEGA v.7.0 [67] was used to analyze the characteristics of the variations in synonymous codon usage, the relative synonymous codon usage values (RSCU), codon usage, and the GC content. 

### 4.3. Repeat Structure and Single Sequence Repeats (SSRs) Analysis

REPuter [68] was used to visualize the various types of repeats in the *Rhipsalis baccifera* cp genome (forward, palindrome, reverse, and complement sequences) with a minimum repeat size of 30 bp and sequence identity of no less than 90% (hamming distance equal to 3). Simple sequence repeats (SSRs) were identified using the software MISA [69] with the following minimum number of repeats: 10 for mono, 5 for di-, 4 for tri-, and 3 for tetra-, 3 for penta, and 3 for hexa-nucleotide SSRs.

### 4.4. Genome Comparison and Sequence Divergence

Comparison of whole chloroplast genomes of eight species, which include; *Talinum paniculatum*, *Portulaca oleraceae*, *Carnegieae gigantea*, *Lophocerus schottii*, *Mammilaria albiflora*, *M. solisioides* and *M. zephyranthoides* (the three species of genus *Mammillaria* represent the cp genome structure of the three structures (1–3) respectively found in its members) was performed and visualized using mVISTA software [70] with the annotation of *Rhipsalis baccifera* as a reference. MAUVE v1.1.1 software [71] was used with its default settings to perform structural genomic sequence alignment of the complete genomes, to study, and analyze structural changes, gene order, and rearrangements. 

### 4.5. Phylogenetic Analysis

Analysis of the phylogenetic relationships and placement of species of suborder Portulacineae was done using complete chloroplast genome sequences. We downloaded 36 previously sequenced members of order Caryophyllales and two additional outgroups from order Lamiales, from the NCBI database (Appendix A) for analysis. Multiple sequence alignment of the 38 complete cp genome sequences was done using MAFFT with default parameters. Model Finder program [72] integrated in phylosuite was used to select the best fit model. GTR+F+G4 model was determined as the best fit substitution model. Phylogenetic reconstructions were first done using the maximum likelihood (ML) method, performed by IQ-Tree that is integrated in the Phylosuite [73] a GUI-based software written in python 3.6.7, and the analyses was run with 1000 bootstrap replicates (Appendix A). Bayesian Inference phylogenies were then inferred using MrBayes 3.2.6 [74] under GTR+G+F model (2 parallel runs, 10,000 generations), in which the initial 25 sampled data were discarded as burn-in (Appendix A). FigTree v1.4.4 was used to visualize and refine the trees. The trees were then combined using AI software (Appendix A).

## 5. Conclusions 

Generally, comparative analysis of the structure of the chloroplast genomes of these selected suborder Portulacineae species has revealed several divergences among their genomes. Most species displayed the typical structure of the cp genomes of order Caryophyllales, with a few species revealing some unique features. Generally, *R. baccifera* and *Mammillaria zephyranthoides,* seem to have a higher number of rearrangements among these species.

We hereby conclude that *R. baccifera* underwent a larger number of mutational events during its evolution, as evidenced by the large number of rearrangements in its cp genome structure. This might have enabled *R. baccifera* to have a high distribution and be able to survive in different environments. We recommend further cp genome analysis of all subspecies of *R. baccifera*, especially those found outside the New World, to analyze and compare their cp genome structure and provide more insight on their relationship.

## Figures and Tables

**Figure 1 plants-09-00979-f001:**
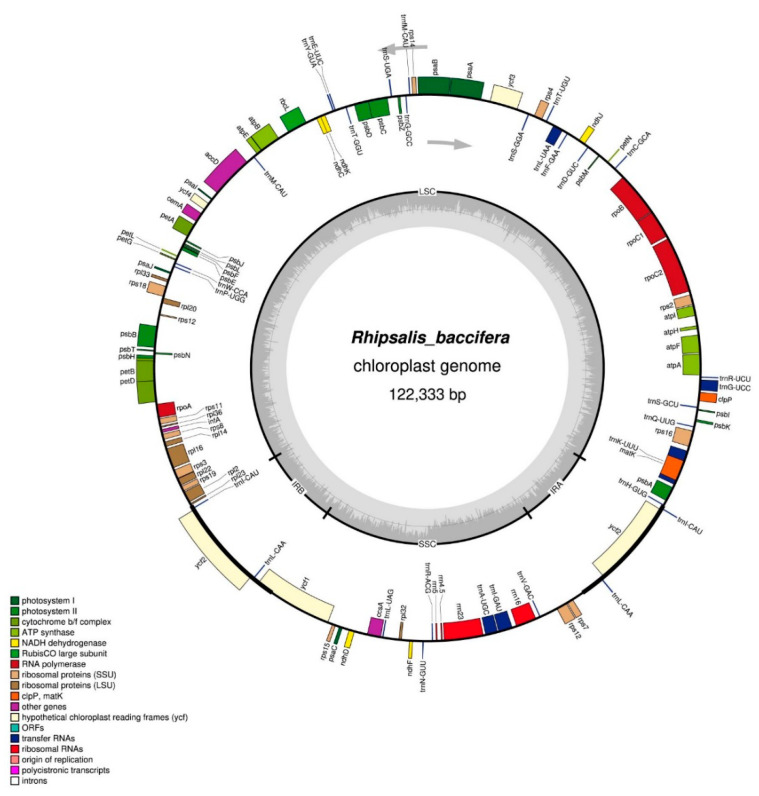
Complete chloroplast genome of *Rhipsalis baccifera*. Genes belonging to different functional groups are shown in different colors. The arrows in the figure represent the transcription direction of genes.

**Figure 2 plants-09-00979-f002:**
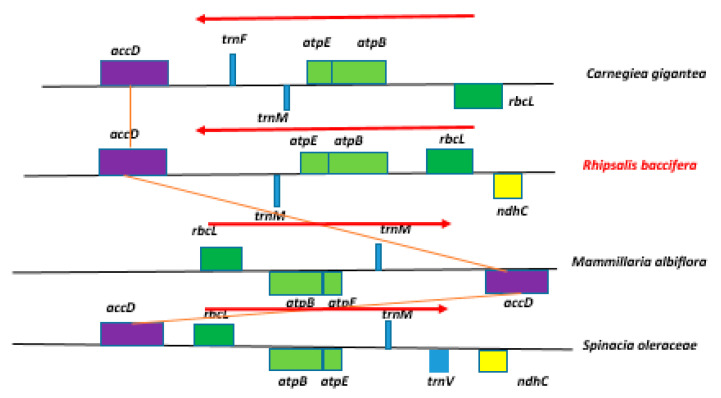
Structural organization of a portion of the LSC region showing location of the ~6kb inversion in the cp genome of *R. baccifera* and other species of suborder Portulacinae relative to *Spinacia oleraceae* cp genome. *Talinum paniculatum* and *Portulaca oleraceae* have similar arrangement of these genes as in *S. oleraceae*.

**Figure 3 plants-09-00979-f003:**
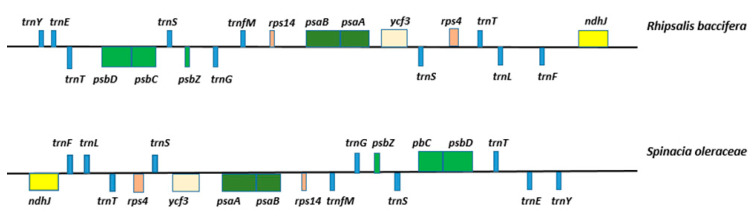
Structural alignment of the LSC region between genes *ndhJ- trnY.* Figure showing inversion in the cp genome of *R. baccifera* relative to *Spinacia oleraceae.*

**Figure 4 plants-09-00979-f004:**
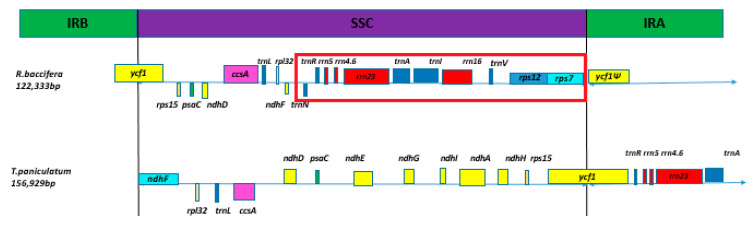
Structural comparison of the SSC region of *R. baccifera* to *Talinum paniculatum*, displaying the SSC inversion in *R. baccifera*. A display of the SSC inversion in *R. baccifera.*

**Figure 5 plants-09-00979-f005:**
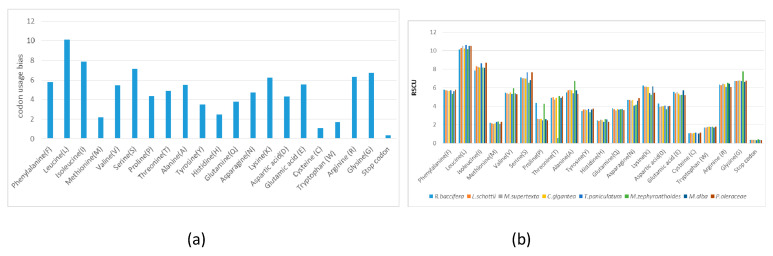
Amino acid composition of proteins coded by the chloroplast genome of *R. baccifera* (**a**) and members of the Portulacineae (**b**).

**Figure 6 plants-09-00979-f006:**
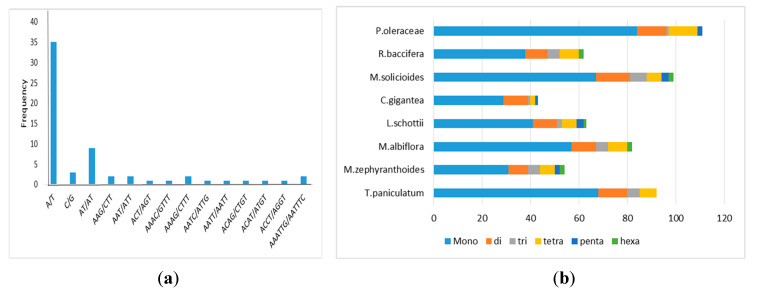
Simple sequence repeats (SSRs) type, distribution, and presence in suborder Portulacineae; (**a**) Number of detected SSR motifs in different repeat types in *R.baccifera* chloroplast genome. (**b**) Number of identified repeat sequences in eight chloroplast genomes.

**Figure 7 plants-09-00979-f007:**
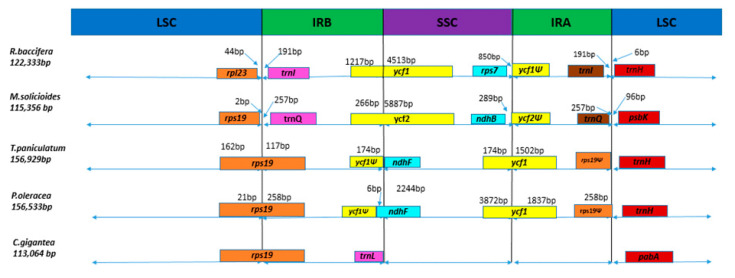
IR borders of the suborder Portulacineae species. Boundaries of the LSC- IRB and the SSC-IRA regions in suborder Portulacineae species. Genes with the symbol Ψ are pseudogenes.

**Figure 8 plants-09-00979-f008:**
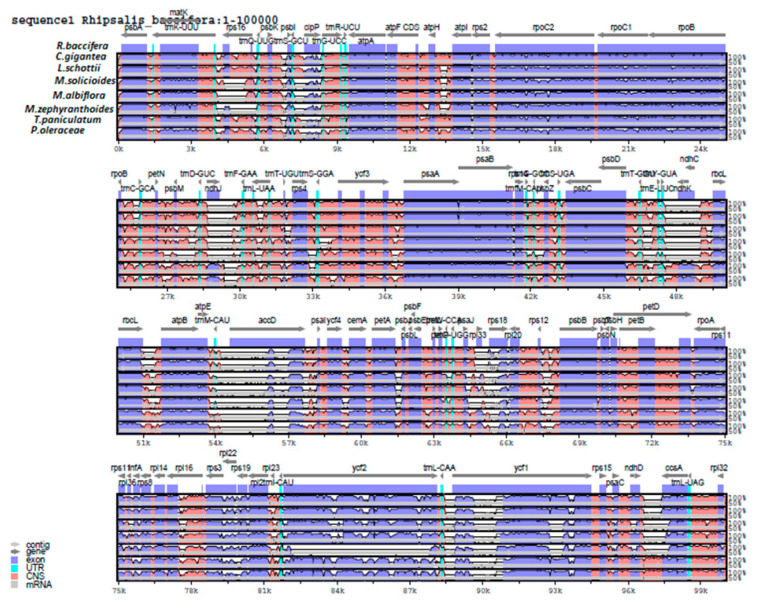
Sequence alignment of eight cp genomes of the suborder Portulacineae using mVISTA program, with the annotation of *R. baccifera* as a reference. The top line shows genes in order (transcriptional direction indicated with arrow). A cut-off of 70% identity was used for the plots, and the Y-scale represents the percent identity between 50–100%. Genome regions are color coded as exon, intron, and conserved non-coding sequences (CNS).

**Figure 9 plants-09-00979-f009:**
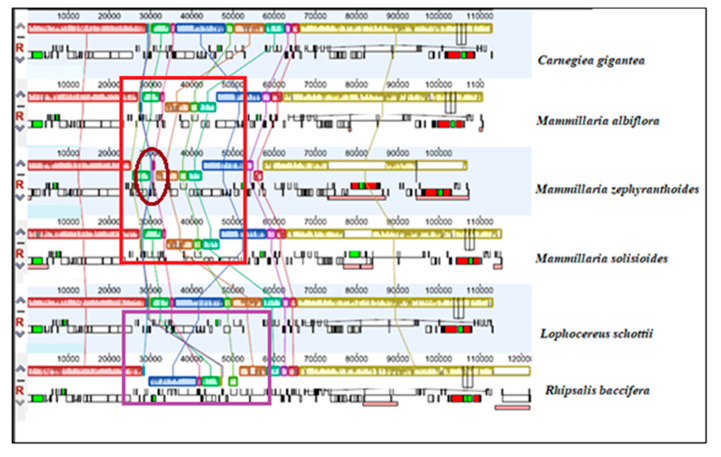
Genome structural alignment map of six chloroplast genomes of species in family Cactaceae, with *Rhipsalis baccifera* as the reference species aligned using MAUVE software. Local collinear blocks within each alignment are represented as blocks of similar color connected with lines. Annotations of *rRNA*, protein coding and *tRNA* genes are shown in red, white, and green boxes, respectively.

**Figure 10 plants-09-00979-f010:**
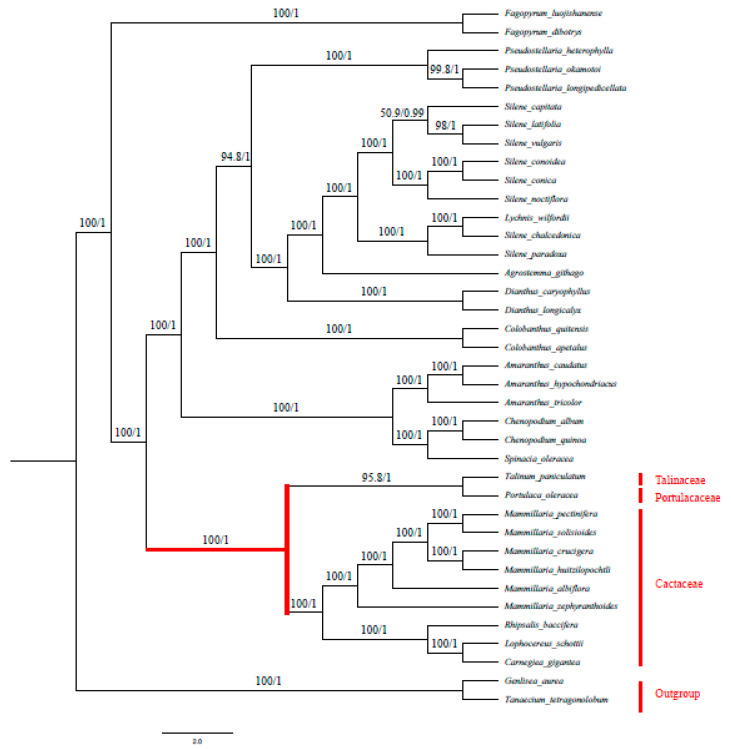
Combined phylogenetic tree of 36 species order Caryophyllales using maximum likelihood (ML) and Bayesian methods based on their complete chloroplast genomes. Two outgroups from order Lamiales were used. The numbers above the nodes are support values with ML bootstrap values on the left and Bayesian posterior probabilities values on the right.

**Table 1 plants-09-00979-t001:** Chloroplast genome composition of *Rhipsalis baccifera*.

Region	Size (bp)	T (U) (%)	A (%)	G (%)	C (%)	Genes
LSC	81,459	32.6	31.4	17.7	18.3	84
SSC	23,531	30.5	30.2	20.3	19.0	19
IRA	8530	31.4	31.7	19.0	17.9	4
IRB	8530	31.4	31.7	19.0	17.9	4
Total	122,333	32.1	31.2	18.3	18.4	110

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
