# Peer review of "Complete Chloroplast Genome of Rhipsalis baccifera, the only Cactus with Natural Distribution in the Old World: Genome Rearrangement, Intron Gain and Loss, and Implications for Phylogenetic Studies"

_plants, 2020, doi:10.3390/plants9080979_

Round 1

Reviewer 1 Report

This manuscript is interesting because it concerns an interesting species of cactus. The work is generally well thought out, well designed and well presented. I have some comments about the selection of samples for analysis, and the general (in my opinion) mess at work. Different species composition appears in various analyzes and figures. It is very difficult to receive work because it is not entirely clear why the analyzed composition of the species has changed. The authors do not explain it to the reader. Suddenly they compare the obtained genome to some species in the engraving and there is no word about this species in the text or in the materials and methods section. There is no information in the results about the type of phylogenetic tree.
There are a lot of typing errors. I have the impression that the authors were in a hurry and ran out of time to refine this manuscript. Bold text appear in strange places and cut off half-sentences. Errors in numbering figures. Some phrases are unfortunate, for example: We represent the first complete chloroplast 19 genome analysis ... ". Genre support is not available for genre names. Lots of errors like: Error! Reference source not found. Some figures require drastic improvement in quality.

I am concerned about the sentence regarding manual correction in the M&M section, i.e. The assembled genome was then manually corrected. What does manual correction mean? What has been improved? How and why?
I also wonder if there will be any errors in this genome. Since there is no GenBank accession number, the administrator can find any errors that will affect, for example, the length of the genome or the number of genes. Of course, I do not question the authors' skills. I assume that everything is fine, but as long as there is no number it can be different.

I think that the number of species also needs clarification, e.g. why eight to compare genomes? What species are these? Why exactly these? And it would be nice to give their names, accession numbers etc. to identify the data used for analysis.

And finally. In my opinion, the conclusions require reflection. Some of these sentences in this section, in my opinion, are a repetition of the results and not conclusions. I also wonder if the title Conclusion and recommendation is admissible by the Editorial Board.

I included my other comments in a pdf file.

Author Response

We are grateful for the great sacrifice of your time and effort to provide these insightful comments to this manuscript. Below is an attached file with our responses. Thank you.

Reviewer 2 Report

The article needs intensive editing, as the English language, so the structure and meaning. Some minor corrections are made in the attached file.

Duplications should be avoided in the Results and Discussion sections.

The use of the terms “chloroplast genome” or “cp genome” in the text should be unified.

Lines 38-39 “The relationship of the members of this clade, however, remains unclear [4-7].”

In fact, the phylogenetic tree in [5] reliably enough reconstructed evolutionary relationships within the ACPT clade.

Lines 71-74 These theses, well-known to schoolchildren, are hardly relevant in the journal issue on molecular phylogenetics.

Table S3 Neither in the text, nor the explanation to the table is the decoding of designations of the type of repeats

Figure 2, corresponding text, and paragraph “Codon usage analysis” Lines 326-338 in the Discussion are not codon usage analysis, but data on amino acid composition of coded proteins. Must be rewritten with figure replaced by a table.

Lines 228-237 It is advisable to give a link to the Table S1 in the Phylogenetic Analysis section and Figure 10 legend.

Lines 231-32 “Rhipsalis baccifera forms a SISTER CLADE to all the other Cactaceae species.”

Clade is a group of species.

Figure 10 legend: “Species of suborder Portulacineae are highlighted in a red selection.”

There is no red selection in the Figure.

Lines 257-58 rps12 gene is generally highly conserved and changes in its structure have been previously suggested to be as A RESULT OF EVOLUTION.

Lines 272-73 “This loss of introns of the petB gene of some Cactaceae species might be as a RESULT OF EVOLUTION.”

Lines 377-378 "We support previous studies that suggest a correlation between changes in the IR  (contraction in this case) and evolution of genomes, consequentially suggesting that the IR  contraction observed in the cp genome of R. baccifera is as a RESULT OF GENOME EVOLUTION.”

And what besides evolution could lead to such results?

Lines 26-28 “High number of rearrangements together with the phylogenetic analysis suggested a LONGER HISTORY OF EVOLUTION for this species.”

Lines 265-67 It is not clear why the non-ancestral state of some features of trnY-psaI and clpP

Is evidence that “R. baccifera undergone a LONGER HISTORY OF EVOLUTION”

Lines 303-305 “We suppose this to be as a result of a LONGER INDEPENDENT HISTORY OF EVOLUTION of this species compared to other members of the order.”

Line 320 “Thus supporting our previous suggestion of R. baccifera having undergone a LONGER EVOLUTION.”

Lines 407-08 “R. baccifera might have undergone A LONGER INDEPENDENT HISTORY OF EVOLUTION compared  to other Cacti species”

Lines 411-12: “R. baccifera has undergone a LONGER HISTORY OF EVOLUTION and our study strongly supports this proposal.”

Line 481:” the LONGER HISTORY OF EVOLUTION in R. baccifera”

The evolution of R. baccifera lasted precisely the same time as the representatives of the sister clade - other cacti.

Line 421 There is no Figure 12 in the paper.

Lines 384-387 “The genes ndhK and ndhJ are absent in the chloroplast genomes of the so far analyzed Cactaceae species except in R. baccifera, while the gene rps16 is pseudogenized in genus Mammillaria, thus THE DIVERGENCE OBSERVED IN THESE GENES.”

Please check this phrase.

Lines 387-92 “The pseudogenization of the rps16 gene has been observed in other studies too, and this has been assumed to be as a result of the dual targeting of the nuclear rps16 copy to the plastid and mitochondria, suggesting that the chloroplast-encoded rps16 gene, which is important for plant survival, has been functionally replaced by a nuclear gene that can encode both mitochondrial and chloroplast thus the chloroplast gene rps16 has been silenced and has become a pseudogene by the nuclear-encoded rps16 [59].”

The too complicated phrase, please check it.

Lines 395-403 “Moreover, the comparative analysis using Mauve reveals inversions in R.baccifera, in the LSC regions; (1) rbcL-trnM and (2) ndhJ-trnY (which might be the most unique feature of this genome) and  (3) a transposition and loss of introns of gene clpP and (4) loss of one intron of rps12 gene. Mammillaria 397 zephyranthoides has two additional rearrangements despite those observed in other species of this

genus i.e., (1) region between genes rps18-trnW and (2) PsbZ-trnT genes. We therefore conclude that, these multiple inversions, intron loss and rearrangements observed in these species is as a result of  parallel evolution in this order, and we agree with previous studies which suggested that there is a correlation between gene rearrangements, and changes in the genome structure of a chloroplast genome of a species and evolution [30, 53,54].”

Very hard for understanding phrase, please check it.

Lines 409-410 “…the unusual long branch leading to the this species when phylogeny is inferred based on chloroplast sequences”

The phylogenetic tree in Figure 10 is a cladogram in which the lengths of the branches are not represented. Therefore, it is difficult to judge how true this statement is.

Lines 317-18 “…the sequence was first the inversion and transposition of the region psaI-petA, then the transposition of the accD-trnM, followed by another…”

Check this phrase

Author Response

We sincerely appreciate the time and effort that you put to provide these very useful corrections. Below is an attached file with our responses. Thank you

Round 2

Reviewer 1 Report

This version of the manuscript is much better. The authors significantly improved their MS. I am also pleased with the authors' responses to my comments. Please check the logical meaning of the sentence from lines 89 to 94. Now it is unclear. I recommend acceptance.

Author Response

Dear reviewer, 

We appreciate the useful suggestions you provided to us in modifying our manuscript and also for recommending this work for acceptance. We have corrected lines 89-94 in the manuscript as suggested. The attached document shows our response.

Thank you.

Reviewer 2 Report

Dear authors,

You presented an interesting study, however further corrections are needed.

Please check also marked yellow points in the attached file.

Lines 29-30, 279

“sister clade”

I must notice again that “Clade” is a group of species, not single species

Line 31

“longer history of evolution for this species”

Lines 472-73

“longer independent history of evolution”

Lines 477, 552

“longer history of evolution”

I must notice again that  evolution of R. baccifera lasted precisely the same time as the representatives of the sister clade - other cacti.

Lines192-202

Figure 2 and corresponding text in the paragraph “Codon usage analysis” are not codon usage analysis, but data on amino acid composition of coded proteins.

Lines 324-326

“This loss of introns of the petB gene of some Cactaceae species might be as a result of evolution and also because petB gene is an important gene in photosynthesis and R.baccifera is an epiphyte , this  loss has also been observed in asterids [25].”

You still use the words "evolutionary" as opposed to the loss of an intron to the loss of a gene. Again, I point out that any observed changes in the genomes of species are the result of evolution.

In addition, epiphytes are not parasites on the supporting plants and require genes for photosynthesis. The article [25] you cited does not mention either the petB gene or epiphytes.

Lines 354

The original text was better.

Lines 357-360

“We suppose that R.baccifera might have undergone some divergent evolutionary history which has enabled it to occur as different subspecies and also the only cactus to occur naturally in the old world. This feature might not be found in all populations of this species and we suggest future work should be done to clarify this”

I recommend deleting the insert you made. This assumption is too hypothetical.

Lines 376-377

“High number of rearrangements have been corresponded to evolution”

Meaningless phrase

Lines 399-400

According to ref.53:

“The most highly rearranged plastomes contain a high frequency of large (>100 bp) repeats”.

Cite [53] more carefully. Authors of [53] state only:

“Trachelium has one of the most highly rearranged chloroplast genomes of land plants and its bizarre organization is clearly associated with the high incidence of dispersed repetitive DNA”

“Short dispersed repeats have been associated with inversion endpoints and occur in a number of taxa”

Lines 401-402

The cited reference [54] does not contain the statement that

 “ there is a positive correlation between the number of repeats

and the degree of rearrangements displayed in a plastome”.

Lines 437-38

“IR contraction observed in the cp genome of R. baccifera is as a result of genome evolution”

Meaningless phrase

Lines 438-441

Refs [35,57] –do not mention PPR4.

Lines 462-468

“The high number of rearrangements, multiple inversions and intron loss in these two species is however not as a result of convergent evolution. They seem to have undergone parallel evolution. We agree with previous studies which suggested that there is a correlation between gene rearrangements, and changes in the genome structure of a chloroplast genome of a species and evolution [30, 53, 54].”

This case is certainly an example of parallel evolution, not convergence. The original text was better.

Ref [30, 54] contain no suggestions on correlation of gene rearrangements with “genome structure of a chloroplast genome of a species and evolution”

Line 475

“the unusual long branch leading to this species when phylogeny is inferred based on chloroplast sequences”.

I am noticing again that he phylogenetic tree in Figure 10 is a cladogram in which the lengths of the branches are not represented. Therefore, it is difficult to judge how true this statement is. You must either delete this comment, or present a tree with branch lengths, or add a table with distances between species under consideration.

Author Response

Dear reviewer,

We appreciate the sacrifice of time and efforts to work on our manuscript and provide these insightful suggestions and corrections. We have made corrections as advised. Below is an attached document with our responses.

Thank you
